# β-Amyloids and Immune Responses Associated with Alzheimer’s Disease

**DOI:** 10.3390/cells13191624

**Published:** 2024-09-28

**Authors:** Elizaveta Kolobova, Irina Petrushanko, Vladimir Mitkevich, Alexander A Makarov, Irina L Grigorova

**Affiliations:** 1Engelhardt Institute of Molecular Biology, Russian Academy of Sciences, 119991 Moscow, Russia; kolobova98@yandex.ru (E.K.); irina-pva@mail.ru (I.P.); mitkevich@gmail.com (V.M.); aamakarov@eimb.ru (A.A.M.); 2Institute of Translational Medicine, Pirogov Russian National Research Medical University, 117513 Moscow, Russia

**Keywords:** Alzheimer’s disease, β-amyloid, post-translational modifications of β-amyloid, immune response, B cells, antibody

## Abstract

Alzheimer’s disease (AD) is associated with the accumulation of β-amyloids (Aβs) and the formation of Aβ plaques in the brain. Various structural forms and isoforms of Aβs that have variable propensities for oligomerization and toxicity and may differentially affect the development of AD have been identified. In addition, there is evidence that β-amyloids are engaged in complex interactions with the innate and adaptive immune systems, both of which may also play a role in the regulation of AD onset and progression. In this review, we discuss what is currently known about the intricate interplay between β-amyloids and the immune response to Aβs with a more in-depth focus on the possible roles of B cells in the pathogenesis of AD.

## 1. Introduction

Alzheimer’s disease (AD) is a neurodegenerative disease characterized by progressive neuronal death and cerebral atrophy, and, as of today, is the most common neurodegenerative pathology [1]. The two major factors associated with AD are senile plaques formed from β-amyloids (Aβs) and neurofibrillary tangles formed by hyperphosphorylated tau proteins accumulating in AD patients’ brains. Several different models (such as the amyloid hypothesis, tau hyperphosphorylation, chronic inflammation, genetic theories, the microbiotic theory, etc.) aimed at explaining the onset of AD have been proposed (discussed in detail in reviews [2,3,4,5,6,7,8,9]). Currently, the amyloid cascade hypothesis, which suggests that the abnormal accumulation of Aβ may be the main trigger of the disease, is the most popular [10]. AD is currently viewed not as an isolated pathology of the central nervous system but as a systemic disease with important roles being played by both the innate and adaptive immune systems, as has been suggested by multiple lines of evidence [11,12,13,14,15,16,17,18,19]. However, the interactions between one of the main actors of the progression of AD, Aβ, and the innate and adaptive immune responses, as well as the effect their interplay has on the development and progression of the disease, are not fully understood.

While the accumulation of extracellular Aβ plaques in the brain is characteristic of AD, Aβ can exist in various structural forms (e.g., monomers, oligomers, and fibrillar structures), which differ in solubility and can have different effects on the pathogenesis of AD [20,21,22]. In addition, there are some mutations [23,24,25,26,27,28,29] and post-translational modifications of the Aβ N-terminus that could affect the rate of pathology development (Table 1) [30,31,32,33,34,35,36,37,38,39,40,41,42,43,44,45,46,47,48,49]. Various forms of Aβ are liable to differ in their propensity for triggering neurotoxicity and neuroinflammation, both of which may affect the accumulation of Aβ in the brain [31,35,39,40,41,43,45,46]. Accumulating Aβ can also trigger an adaptive immune response by T cells and B cells, and an autoantibody response that may in turn affect the progression of the disease (Figure 1) [17,50,51,52,53,54]. However, the propensity of various Aβ structures to trigger immune responses that may in turn promote or impede AD development and associated pathology is poorly understood. In this review, we highlight what is currently known about various forms of Aβ and their ability to trigger pathology, as well as immune responses. We also provide more in-depth analysis of the B cells’ role in the regulation of AD progression. Based on an analysis of the published data, we attempt to outline the gaps in the current understanding of the Aβ–immune system interactions and their role in AD.

The accumulation of various forms of β-amyloid in the brain parenchyma leads to the activation of resident innate immune cells (astrocytes and microglia) that then promote chronic neuroinflammation and AD-associated pathology. This neuroinflammation can be either further augmented by B cells that secrete pro-inflammatory interleukins (e.g., IL-6, IL-12, IL-15), or suppressed by regulatory B cells that produce anti-inflammatory factors (e.g., IL-10 and IL-35). IL-35 may also inhibit the expression of β-secretase, thereby limiting the production of Aβ from APP and accumulation of β-amyloids. Neuroinflammation can be also positively or negatively regulated by T cells that gain access into the brain parenchyma.

In AD patients, Aβ can induce T cell responses that may be influenced by B cells in antigen-dependent or independent ways. Different forms of β-amyloid may drain from the brain into the SLO, where they may be acquired by dendritic cells (DCs) that could stimulate Aβ-specific T cells that recognize MHC/Aβ peptide complexes presented on DCs via their T cell receptors (TCRs). β-amyloids also activate Aβ-specific B cells by interactions with their B cell receptors (BCRs). Activated B cells engulf BCR/antigen complexes, proteolyze them, and, similarly to DCs, can present MHCII/peptide complexes for specific recognition by helper T cells. These interactions may promote the stimulation of Aβ-specific helper T cells and acquisition of T cell help by the antigen-presenting B cells leading to the generation of Aβ-specific memory B cells and long-lived plasma cells.

Aβ-specific autoantibodies secreted by plasma cells may access the brain parenchyma directly and/or interfere with β-amyloid peptide transport from the blood and lymphatic systems into the brain. In these ways, Aβ-specific antibodies may qualitatively and quantitatively alter the composition of β-amyloid deposits in the brain and possibly modulate the progression of AD.

## 2. The Role of β-Amyloid and Its Forms in the Pathogenesis of AD

The molecular basis for the occurrence of AD is currently ambiguous. However, the disease is associated with the accumulation of Aβ fibrillary plaques consisting of Aβ peptides [55]. Aβ peptides are formed by the β- and γ-secretase-dependent endoproteolytic cleavage of amyloid-β precursor protein (APP), which is expressed in neurons located in the brain, glial cells, endothelial cells, and platelets [56]. The length of the resulting peptides varies from 37 to 43 amino acid residues [55,57,58], but the predominating forms are Aβ_1–40_ and Aβ_1–42_ [59,60]. Although Aβ is a normal component of biological fluids (in picomolar concentrations) [61], abnormalities in Aβ accumulation and aggregation are associated with AD. For example, mutations in the *PSEN1* (presenilin-1) and *PSEN2* (presenilin-2) genes alter the proteolytic cleavage of APP, thereby provoking the accumulation of Aβ_1–42_, a peptide more prone to aggregation than Aβ_1–40_ [62]. In addition, specific mutations in the genes encoding APP are a risk factor for the development of AD at young to middle ages, likely due to the increased propensity of Aβ mutants to aggregate [24,26,27] (Table 1).

**Table 1 cells-13-01624-t001:** The most frequent/well-studied Aβ mutations and post-translational modifications in the metal-binding region (N-terminus) of Aβ that affect its oligomerization and aggregation.

Mutation in Aβ	Effects
A2V	Increased Aβ production [63,64] Acceleration of Aβ aggregation [64,65,66]
English (H6R)	Increased Zn^2+^-dependent oligomerization of Aβ [23] Increased neurotoxicity of Aβ oligomers [24,25] Acceleration of Aβ fibril formation [26]
Tottori (D7N)	Acceleration of Aβ aggregation [27] Increased oligomerization of Aβ [24] Increased neurotoxicity of Aβ oligomers [24] Acceleration of Aβ fibril formation [26]
Taiwan (D7H)	Increased Aβ production [28,29] Increased Aβ42/40 ratio [28] Increased Zn^2+^/Cu^2+^-dependent oligomerization of Aβ [28,29] Increased neurotoxicity of Aβ oligomers [28,29]
**Post-translational modification of Aβ**	**Effects**
Iso-D7-Aβ (isomerized Asp7 residue)	Increased Zn^2+^-dependent oligomerization of Aβ [34,36,38,43,48,49] Increased neurotoxicity of Aβ oligomers [43] Age-related accumulation [44]
3pE-Aβ(pyroglutamate modified glutamic acid residue)	Acceleration of Aβ aggregation [31,46] Increased oligomerization of Aβ [31,45,47] Increased neurotoxicity of Aβ oligomers [31,35,45,46] Age-related accumulation [32]
pS8-Aβ(phosphorylated Ser8 residue)	Acceleration of Aβ aggregation [39,40,41] Increased oligomerization of Aβ [39,40,41] Increased neurotoxicity of Aβ oligomers [39,40,41] Age-related accumulation in APP-PS1 tg mice [39,40]
	Inhibition of Zn^2+^-dependent oligomerization of Aβ [30] Prevention of Na^+^/K^+^ ATPase inhibition [30] Reduction of amyloid plaques in APP-PS1 tg mice [30,37]

Excessive accumulation of β-amyloid in the brains of AD patients may be caused by increased production of Aβ. Alternatively, Aβ buildup might occur due to an insufficiently rapid rate of its clearance from the extracellular space by phagocytosis (mediated by astrocytes and glia cells) and degradation (by proteases secreted by astrocytes) [14,67,68]. Another process affecting Aβ brain concentration is its transportation between the brain, bloodstream, and lymphatic system. Any change in the dynamic equilibrium maintained by the production, removal, and transportation of Aβ can lead to its accumulation and aggregation [69,70] (Figure 1). However, the extent to which each of the above processes influences the development of AD remains unclear.

Aβ is present in mammals in various forms, including the monomeric, dimeric, oligomeric, and fibrillar structures, which contribute in various ways to the pathology leading to AD. For an extended period, AD research has focused on senile plaques in the brain that mainly consist of insoluble Aβ aggregates in the extracellular space. The formation and accumulation of plaques was originally thought to cause vascular damage, neuronal cell death, and progression of dementia [71]. However, neuronal death has been observed in patients’ brain regions devoid of plaques, suggesting no direct correlation between the manifestation of the disease and the number of plaques. Moreover, senile plaques have been found in people without clear manifestations of cognitive impairment. These findings suggest that insoluble Aβ aggregates are not likely to be the main cause for neurodegeneration [20]. More recent studies have linked cognitive impairment to the presence of large amounts of soluble Aβ oligomers [21,22], the most toxic of Aβ species, which leads to cell necrosis or apoptosis through various mechanisms.

### 2.1. Possible Mechanisms of Aβ Oligomer Toxicity

Based on studies comparing the effects of various types of Aβ aggregates on cellular membranes (reviewed in [72]), Aβ oligomers are generally considered to be more cytotoxic than fibrils. Soluble Aβ oligomers may contribute to neuronal death in a number of different ways including the disruption of cellular membranes, dysregulation of ion flows, pathological engagement of membrane receptors, induction of oxidative stress, and AD-related target protein phosphorylation [73]. Importantly, post-translational modifications of Aβ (Table 1) can significantly alter its cytotoxic effects [33].

#### 2.1.1. Disruption of Cellular Membranes

Under certain conditions, Aβ oligomers can form pores in cellular membranes, thereby disrupting their ion permeability [74]. Some studies suggest that Aβ monomers and dimers can directly insert themselves into membranes, where they can form toxic pores by interacting with Aβ oligomers in the membrane [75]. In this respect, it is important to take note of the lipid-chaperone hypothesis (LCH), which suggests that the transport of intrinsically disordered proteins (IDPs) from the aqueous phase into the lipid bilayer may be facilitated by their binding to free phospholipids in the aqueous phase [76]. The LCH model suggests that lipid dysregulation may be contributing to the formation of toxic pores by Aβ peptides, as well as other IDPs (including islet amyloid polypeptides (IAPPs), a-synuclein, amylin, and β-synuclein) that are associated with various human diseases [77,78,79]. While the LCH hypothesis requires further experimental validation, taken that the ɛ4 allele of apolipoprotein E (the main extracellular lipid and cholesterol transporter in the brain that could also bind Aβ [80]) is a prominent genetic risk factor for AD [81], the possible role of lipid association with Aβ and oligomers should not be disregarded.

#### 2.1.2. Dysregulation of Ion Flows

Disruption in ion homeostasis is one of the key factors in AD [82,83,84]. Impaired Na^+^/K^+^ [85,86,87,88] and calcium homeostasis [89] in AD has been observed not only in neurons, but also in other cell types [90]. The disturbance of ion homeostasis in Aβ-exposed cells may be caused by impaired membrane barrier properties (as discussed above) and/or due to the direct interaction of Aβ with ion-transporting proteins and receptors regulating ion entry (reviewed in [91]). Aβ has been shown to directly bind and inhibit Na^+^/K^+^ ATPases, leading to an impaired sodium–potassium gradient in neuronal cells [85,87,92,93]. This results in the disruption of normal neuronal activity, calcium overload, and cell death [85,87,88,94,95]. In addition, Aβs promote the activation of N-methyl-D-aspartate (NMDA) receptors leading to calcium entry and an increase in its intracellular concentration [96]. Aβ oligomers have also been shown to increase mitochondrial Ca^2+^ [97], thereby promoting mitochondrial dysfunction, cell death [98], and synaptic failure (reviewed in [99]).

#### 2.1.3. Pathological Engagement of Membrane Receptors

Some of the cytotoxic effects occur as a consequence of Aβ interaction with multiple membrane receptors [100], including the NR2B-containing NMDA receptor [101]; α7-nicotinic acetylcholine receptor [102,103]; PrP^c^ receptor, that may regulate the synaptic function of hippocampal neurons [104,105,106]; the receptor for advanced glycation end products (RAGE), which carries Aβ from the blood to the brain across the blood–brain barrier [107]; and other receptors [100].

These receptors can bind monomeric, oligomeric, and fibrillar forms of Aβ [100]. Moreover, both Aβ monomers and oligomers can interact with some receptors [100], likely contributing to both their physiological and pathological functions. Another possibility is that Aβ oligomerization occurs following its binding to surface receptors, as has been proposed for Aβ interactions with Na^+^/K^+^ ATPases [30]. The oligomerization of Aβ seeded on Na^+^/K^+^ ATPases leads to the inhibition of the enzyme’s activity, thereby disrupting the normal functioning of neurons [85,93].

Importantly, the interactions of Aβ isoforms containing some post-translational modifications with membrane receptors differ from those of its unmodified form, which leads to increased neurotoxicity [108,109] (Table 1).

#### 2.1.4. Induction of Oxidative Stress

In addition to the scenarios described above, Aβ oligomers have been shown to promote cytotoxicity by inducing oxidative stress in neuronal cells, leading to, among other things, lipid peroxidation and the disruption of cell membrane barrier properties [110,111]. The precise molecular mechanism of oxidative stress induction in neurons by Aβ is unclear. However, it is at least partially mediated by decreased intracellular glutathione, which is the main thiol and cellular antioxidant [112,113,114]. Alterations in intracellular redox status lead to redox-dependent modifications of proteins (nitrosylation and glutathionylation) and changes in their functions [115,116,117]. For example, increased nitrosylation of cyclin-dependent kinase 5 (Cdk5) that is associated with AD promotes Cdk5 activation and contributes to NMDAR-mediated neuronal dendritic spine loss that is induced by Aβ [118]. Importantly, post-translational modifications of Aβ can significantly alter the effect that Aβ has on the redox status of cells [43,108,119,120] (Table 1).

#### 2.1.5. AD-Related Target Protein Phosphorylation

In addition to the accumulation of Aβ, the hyperphosphorylation of tau proteins that promotes the formation of intracellular neurofibrillary tangles and neuritic dystrophy is characteristic for AD. However, a number of studies suggest that accumulation of the Aβ oligomers precedes the hyperphosphorylation of tau proteins in AD (reviewed in ([10])). Importantly, Aβ_1–42_ oligomers have been shown to induce the increased phosphorylation of several target proteins in cultures of neuronal cell lines (also characteristic for AD), including the hyperphosphorylation of tau proteins and other proteins involved into microtubule (MT) assembly [121,122,123]. While the mechanism of this regulation is not fully understood, Aβ has been shown to alter the activities of phosphatidylinositol-3-kinase, Akt, and protein kinase C (PKC) that, through their downstream target glycogen synthase kinase-3β (GSK3β), may regulate the phosphorylation of tau proteins [124].

Of specific interest is that some common post-translational modifications of Aβ promote the increased phosphorylation of proteins in neuronal cell lines, including the MT-regulation-associated tau protein, tubulins, and matrin 3 [123].

Based on the studies described above, various forms of Aβ can differentially affect mechanisms that regulate cytotoxicity, with Aβ oligomers and some post-translational modifications of Aβ likely pathologically contributing to the development of neurotoxicity associated with AD.

### 2.2. Scenarios of Aβ Oligomerization and Post-Translational Modifications

Aβ oligomers can form in various ways. One possible scenario is for Aβ peptides, which are folded into two parallel β-sheets (residues 12–24 and 30–40), to form oligomers via intermolecular interactions with each other within the 25–29 bended regions, which subsequently assemble into Aβ fibrils and β-fold structures [125,126]. Inside these fibrillar structures, the middle and C-terminus regions of Aβ peptides (that could be tentatively divided into Aβ_12–29_ and Aβ_30–40_) are located within the protein complexes, while their N-termini are exposed [53,126,127]. Other scenarios of Aβ structural oligomerization occur in the presence of transition metal ions (e.g., Zn^2+^, Cu^2+^, Fe^2+^, Al^3+^) and depend on Aβ mutations, post-translational modifications, as well as other factors [29,128,129]. Aβ_1–42_ is among the most aggregation-prone peptides that forms oligomers, and is a major component of amyloid plaques [130,131,132]. In addition, some post-translational modifications of Aβ peptides are known to accelerate their oligomerization (Table 1).

One of the most common post-translational modifications of β-amyloids is Aβ with an isomerized Asp7 residue (Iso-D7-Aβ) [133] (Table 1). The Iso-D7-Aβ form possesses a significantly higher propensity for oligomerization than Aβ_1–42_, and depends on the binding of zinc ions to the Aβ_1–16_ metal-binding domain [34,36,38,43,48,49]. The coordination of Zn^2+^ promotes the dimerization and oligomerization of Aβ peptides via segment Aβ_11–14_ [134]. The Iso-D7-Aβ isoform has been shown to induce greater neurotoxicity in cells compared to intact Aβ_1–42_ [43]. In transgenic nematodes (that express human Aβ), Iso-D7-Aβ induces the zinc-dependent oligomerization of endogenous Aβ, which leads to a significant reduction in animal lifespan [135]. It also accelerates cerebral amyloidosis upon intravenous administration into mouse models of AD [136].

The presence of Iso-D7-Aβ in the brains of AD patients was first described in 1993 [137]. The isoform was detected both in extracellular Aβ deposits and in cerebral blood vessels [133,138]. The amount of Iso-D7-Aβ in the brain was shown to be positively correlated with patients’ ages [44], suggesting either age-dependent increases in the formation of the isomerized form of Aβ [139] or a decrease in its deisomerization, degradation, phagocytosis, or transport. Based on the increased ability of Iso-D7-Aβ to promote Aβ oligomerization and induce neurotoxicity, age-dependent accumulation of Iso-D7-Aβ in the brain and blood may contribute to AD development.

In addition to Iso-D7-Aβ, there are other post-translational modifications within the Aβ_1–16_ metal-binding domain that may influence the development of AD (Table 1). One of the most well-studied is the pyroglutamate-modified form of Aβ (3pE-Aβ), which is truncated of the first two amino-acids. 3pE-Aβ also accumulates in the brain tissue with age [32]. It is generally accepted that 3pE-Aβ accelerates the formation of Aβ oligomers [31,45,47] and Aβ aggregation [31,46], and increases Aβ toxicity to neurons [31,35,45,46] (Table 1). Another post-translational modification that has been reported to exert an effect on Aβ oligomerization is Aβ phosphorylated on the serine 8 residue (pS8-Aβ) (Table 1). There is some evidence that pS8-Aβ increases the oligomerization and aggregation of Aβ, as well as the neurotoxicity of Aβ oligomers [39,40,41]. However, other studies indicate the inhibition of Zn^2+^-dependent oligomerization [30] and reduced amyloid deposition in APP-PS1 transgenic mice upon intravenous administration of pS8-Aβ [37] (Table 1 and Table 2).

In addition to post-translational modifications, the differential activity of α/β/ɣ-secretases and other enzymes can generate shortened forms of Aβ that may also influence AD pathogenesis [147].

While various forms of Aβ accumulate in humans in the process of normal aging and in AD patients, the understanding of their interaction with the innate immune system and ability to promote neuroinflammation and regulate progression of the disease is currently limited.

## 3. The Interaction of Aβ with the Innate Immune System and the Pathogenesis of Alzheimer’s Disease

Neuroinflammation is believed to play a major role in the pathology associated with AD [15,148,149,150]. Protein aggregates, dead neurons, and various pathogens can trigger the innate immune cells of the brain (microglia and astrocytes) to migrate to the site of damage and initiate neuroinflammation [13,14,15,19,151]. Importantly, neuroinflammation may have a dual function by playing a neuroprotective role during the acute phase of the immune response, but contributing to pathological processes during chronic activation of the involved cells [152]. It is possible that chronic activation of microglia and astrocytes is supported by cell death induced by soluble Aβ oligomers and Iso-D7-Aβ or other post-translationally modified forms of Aβ (Figure 1, Table 1). Neutrophils and other phagocytes may also potentially contribute to AD-associated neuroinflammation while regulating Aβ aggregation and accumulation. On one hand, neutrophils could inhibit Aβ aggregation and neurotoxicity by secreting neutrophil granule proteins [153]. On the other hand, they may contribute to oxidative stress that negatively affects Aβ phagocytosis [154,155].

An analysis of brain tissue RNA from AD patients and controls suggested an elevated signature of monocytes, MO macrophages, and dendritic cells and a decreased presence of resting NK cells in AD patients [156]. In addition, reduced prevalence and cytotoxicity of NK cells in AD patients have been reported [157]. In that respect, of interest is the ongoing phase II trial of SNK01 as an AD therapy (where patients’ NK cells are expanded and activated ex vivo), which has shown promising results in phase I trials [158]. Further studies are desirable to better dissect the role of NK cells in the regulation of inflammation and development/progression of AD. In addition, while Aβ has been shown to impact both monocytes and neutrophils, whether Aβ could also exert a direct action on NK cells remains unclear (reviewed in [159]).

Overall, the relationship between neuroinflammation and the accumulation of Aβ in AD is very complex, as in some cases the phagocytosis of Aβ by innate immune cells [160] may also promote their inflammatory response, while in other cases neuroinflammation appears to trigger the accumulation of Aβ [148].

In addition to the well-established role of innate immunity in the pathogenesis of AD, there is evidence suggesting the involvement of the adaptive immune system in regulating the disease. This regulation may depend, at least in part, on the antigen-specific response of T and B lymphocytes to Aβ that has been detected in humans and will be discussed below.

## 4. The Interaction of Aβ with the Adaptive Immune System and the Pathogenesis of Alzheimer’s Disease

Antigens from the brain parenchyma can enter the lymphatic system and, by means of perivascular drainage, the glymphatic system, or lymphatic drainage of cerebrospinal fluid (CSF), reach the deep cervical lymph nodes [161,162], where they can activate antigen-specific T and B cells (Figure 1). Thus, there is a possibility that Aβ peptides and their soluble multimeric forms in the brain may enter cervical lymph nodes and induce the activation of Aβ-specific lymphocytes. An adaptive immune response to Aβ may also take place in other secondary lymphoid organs (SLOs) in the case of more systemic spread of the autoantigens.

### 4.1. Aβ-Specific T Cells in AD

Possible mechanisms of T cell participation in AD onset and progression have been addressed in multiple studies [11,16,17]. T cells with an activated phenotype accumulate in the brain parenchyma of AD patients (as well as in some mouse models of AD) [163,164,165], suggesting their likely contribution to the regulation of AD. The accumulating T cells are enriched in the hippocampal region, where the main histologic changes associated with AD are manifested, including the excessive deposition of β-amyloid and hyperphosphorylation of tau proteins, as well as neuronal death [166]. A recent study also showed an accumulation of T cell receptor (TCR) clonal CD8 effector memory T cells (T_EMRA_) in the CSF of AD patients, some of which had specificity to the Epstein–Barr virus [167]. Locally present effector T cells are likely to promote neuroinflammation in the brain. However, in some cases they may limit neuroinflammation [17,50,51]. In addition, neuroinflammation should be negatively controlled by the regulatory T cells that represent about 10–30% of all CD4+ T cells in a normal human brain [168]. Which subpopulations of T cells play the dominant role in immunoregulation in the brain of patients with AD, and what their antigen specificity is, is still unclear (Figure 1).

Consistent with the Aβ-mediated activation of antigen-specific lymphocytes, increased reactivity to Aβ in CD4 and CD8 T cells derived from AD patients has been reported [169,170,171,172]. One of these studies suggested that Aβ_1–42_ is more immunogenic for human T cells than Aβ_1–40_ and, at the same time, that Aβ_16–30_ epitopes elicit the highest number of T cell clones responding to it [171]. However, the role of Aβ-specific T cells in the pathogenesis of AD in humans is unclear.

Based on studies in various murine models of AD (Table 2), Aβ-specific CD4 T cells can crosstalk with microglia and either promote or impede the development of the disease [50,173]. Aβ-specific T cells generated in mice have been shown to induce both pro-inflammatory activation (for Th1 and Th17 cells) and more regulatory action (for Th2 cells) in cocultured mixed glial cells [51]. When adoptively transferred into older (6–7 months old) APP/PS1 transgenic mice (Table 2), Aβ-specific T cells infiltrated mouse brains, where IFNγ-producing Th1-like T cells promoted the activation of glia and deposition of Aβ, thereby significantly accelerating disease development [174]. Interestingly, the infection of APP/PS1 mice with *Bordetella pertussis* induced significant infiltration of the endogenous IFNγ- and IL-17-producing T cells in the brains of mice, also eliciting glial inflammation and Aβ deposition [175].

However, in another mouse model of AD, Aβ-specific IFNγ-producing Th1 cells facilitated the clearance of Aβ deposits by microglia. In this model, APP-Tg J20 mice (Table 2) were directly immunized with the Aβ_10–24_ peptide in adjuvant. While immunization with the immunodominant Aβ peptide generated a robust Aβ-specific Th1 response and no Aβ-specific Abs, T cells gained access into the brains only in APP Tg mice crossed to SJL mice expressing IFN-γ in the central nervous system (CNS) under a myelin basic protein (MBP) promoter (Table 2). The immunization of 9-month-old APP/IFN-γ double Tg mice led to an accumulation of CD4^+^ T cells and macrophages in the hippocampus, and to some extent in the cortex and cerebellum, resulting in temporary meningoencephalitis, the activation of microglia, and elevated expression of the CD86 costimulatory molecule by microglia at the sites of Aβ plaques [146]. It should be noted that immunization with Aβ1–42 in an adjuvant also resulted in meningoencephalitis in about 6% of AD patients in a human clinical trial [176]. The Th1-induced pro-inflammatory response in the brains of APP/IFN-γ mice lasted for a month and led to the enhanced clearance of Aβ due to glial cell uptake [146]. Similar effects were observed in the 5XFAD mouse model (Table 2), where Aβ-specific CD4^+^ T_H_1 cells abrogated AD-like pathology via a subpopulation of microglia that expressed the major histocompatibility complex class II (MHC II) [50].

Overall, the studies described above suggest that in some mouse models Aβ-specific T cells can access the brain where they interact with local glia and microglia and play a role in the AD-like disease development. However, why IFN-γ-producing Aβ-specific Th1 cells are associated with enhanced Aβ deposition in some mouse models and increased clearance of Aβ plaques in other models is unclear. In addition, whether mouse-model-derived conclusions about Aβ-specific T cells may be relevant to human AD pathology remains an open question.

To summarize all of the above, while elevated levels of Aβ-specific T cells have been detected in the blood and an accumulation of T cells (including some clonally expanded CD8 T cells) was found in the brains and CSF of patients with AD, the presence of Aβ-specific CD4 and CD8 T cells in the brains of AD patients has not been yet reported. Moreover, the function of Aβ-specific T cells in the regulation of brain inflammation, Aβ deposition, as well as cytotoxic responses in human AD patients remains to be established. In addition to the more direct mechanisms of action described above, Aβ-specific CD4^+^ T cells may be involved in providing help to Aβ-specific B cells that may also play an important role in the regulation of AD.

### 4.2. Aβ-Specific and Other B Cells in AD

A number of studies have reported the presence of IgG autoantibodies to Aβ in the blood and CSF of both healthy volunteers and AD patients [127,177,178,179,180]. This indicates the involvement of Aβ-specific B cells in the immune response, during which they differentiate into plasma cells secreting autoantibodies to Aβ (Figure 1). While the activation of Aβ-specific B cells and formation of Aβ-specific autoantibodies may play a role in the onset and progression of AD, their interplay with various structural forms of Aβ in the regulation of AD is ambiguous.

#### 4.2.1. Aβ-Specific Antibodies

One of the important questions for AD diagnostics and the analysis of its regulation is whether AD patients have an elevated or reduced proportion of autoantibodies to Aβ. Interestingly, some works reported reduced titers of IgG to Aβ_1–42_ in AD patients compared to healthy donors [177,179], while no difference was detected in other studies [127,180]. The observed discrepancy may be due to different methods of Aβ-specific antibody detection or differences between the AD groups selected for analysis [127,177,179,180].

As indicated above, β-amyloid can exist in the body in various multimeric forms. All forms may contribute to the pathogenesis of AD and induce a humoral response to different epitopes of β-amyloid formations [127,181]. Antibodies can target Aβ peptides at the N-terminal, the middle, and the C-terminal sites. Additional structural epitopes for antibody binding may arise or become hidden due to Aβ oligomerization and aggregation. For example, the increased formation of fibrillar Aβ may lead to decreased surface exposure of the middle and C-terminus regions of engaged Aβ peptides [127]. Consistent with this hypothesis, one study reported increased autoantibodies to Aβ_1–12_ and decreased Aβ_19–30_- and Aβ_25–36_-specific IgG in the blood and CSF of AD patients compared to healthy donors and patients with preclinical AD. Moreover, increased antibody titers to the N-terminus of Aβ correlated with increased amyloid burden in the brain and cognitive decline in AD patients [127]. The observed increase in autoantibodies to the N-terminus of Aβ likely reflects the accumulation of fibrillar amyloids in AD patients, in which the N-terminus is accessible while other Aβ regions are more buried for B cell activation and the formation of new autoantibodies.

How the availability of autoantibodies to different epitopes of Aβ influences the development of AD has been examined in the PDAPP mouse model of AD [53] (Table 2). PDAPP transgenic mice were immunized with a T cell epitope from ovalbumin bound to either different N-terminal Aβ peptides (Aβ_1–5_, Aβ_3–9_, or Aβ_5–11_), or to a fragment derived from the middle region of the peptide (Aβ_15–24_). The immunization of mice with each of the three N-terminal peptides significantly reduced the amount of amyloid deposits in the brain. In addition, Aβ_3–9_ and Aβ_5–11_ significantly reduced neuronal dystrophy and death. In contrast, immunization with an antigen including Aβ_15–24_ did not reduce amyloid load and did not protect against neuronal dystrophy or death [53]. The length of Aβ_3–9_ and Aβ_5–11_ peptides is insufficient for their presentation in complex with major histocompatibility complexes (MHCI and/or MHCII) and T cell activation. On this basis, it can be hypothesized that in this mouse model of AD, the antibodies to the N-terminus, but not to the mid-domain epitopes, influence the amount of Aβ deposited in the brain [53]. While immunization with complexes that include epitopes of N-terminal Aβ peptides had a therapeutic effect on AD mice [52,53], studies performed on the immunization of AD patients with antigens containing these Aβ epitopes did not have a positive effect on the course of the disease [182,183].

In addition to the studies described above, the functional activity of antibodies to different epitopes of Aβ was investigated for a number of monoclonal antibodies (mAbs) developed for AD therapy (including Bapinezumab, Lecanemab, Crenezumab, Ponezumab, Gantenerumab, and Solanezumab), which can bind to monomeric, oligomeric, or fibrillar forms of Aβ_1–42_ [184]. In studies conducted on mice, these drugs exerted therapeutic effects without causing significant side effects. Moreover, it has been demonstrated that, when passing through the blood–brain barrier (BBB), anti-Aβ mAbs may inhibit the toxic oligomerization of Aβ, promote the phagocytic clearance of microglia plaques in brain parenchyma [185], and/or exert anti-inflammatory effects by reducing the production of pro-inflammatory cytokines [186]. However, testing of these mAbs, including Bapineuzumab (a mAb specific to the Aβ_1–5_ epitope), Lecanemab (a mAb specific to the N-terminus of Aβ), and Solanezumab or Crenezumab (mAbs specific to the middle-domain Aβ_16–26_ or Aβ_13–24_ epitopes) in clinical trials has not led to any significant improvement or very minor positive effects (for Lecanemab) in patients’ conditions. Moreover, some of these drugs (e.g., Bapineuzumab) resulted in life-threatening side effects such as vasogenic edema, microbleeding, and cerebral ischemia [184]. It is believed that the occurrence of such side effects may be caused by the binding of antibodies to β-amyloid deposits in the blood vessels of the brain [187].

There are at least two possible explanations for the modest effects observed after the administration of mAbs to Aβ in humans with AD. First, it is possible that better control of pathogenic Aβ oligomers by mAbs cannot reverse pathological neurodegeneration during the later stages of disease progression (when AD is usually detected in human patients). Second, the difference in the effects observed in mice and humans could be due to more difficulty in antibodies accessing the brain parenchyma from the blood in humans compared to mice, even with impaired permeability of the BBB induced by the development of AD [188]. In that case, it is conceivable that if Aβ-specific plasma cells infiltrate the brain, the local production of antibodies to Aβ could have a greater impact on the course of AD. It should be noted that a postmortem analysis of brain sections showed increased amounts of IgA in the brain parenchyma of AD patients [189]. However, the local presence of plasma cells and specificity of the antibodies in the brain parenchyma of AD patients is still unclear.

It is also unclear whether AD-afflicted and/or normal elderly patients, who often accumulate elevated levels of Iso-D7-Aβ and 3pE-Aβ in the brain, have autoantibodies that specifically recognize these forms of Aβ. The role of the Iso-D7-Aβ-specific mAb with no cross-reactivity to the intact form of Aβ has been examined once on the 5xFAD mouse model of AD [190] (Table 2). Administration of this mAb resulted in a reduction in the total amount of unmodified Aβ in the brains of mice to the same extent as the administration of a control mAb specific to the N-terminus of Aβ. Moreover, in both cases, an improvement in cognitive function was observed in the mice as compared to the control group [139]. In addition, multiple studies have been devoted to the development of antibodies or vaccines selective against 3pE-Aβ. To date, there are monoclonal mouse antibodies (TAP01, 9D5, 3B8, 07/01, BAMB31, mE8) capable of binding both oligomers [191,192,193] and larger deposits of 3pE-Aβ in plaques [192,194]. Some of these antibodies exerted positive effects on cognitive function [191,192,195] and reduced the deposition of amyloids in various mouse models of AD, while causing no significant side effects in mice [191,192,193,194,195,196,197]. Subsequently, a humanized IgG_1_ mAb to 3pE-Aβ (Donanemab) has been developed based on the murine mE8-IgG_2a_. Donanemab has shown positive results in trials in AD patients, especially at the early stages of the disease. Although the safety of this drug has been questionable [198,199], Donanemab was recently approved for use by the United States Food and Drug Administration [200].

Based on the fact that Iso-D7-Aβ and 3pE-Aβ are peculiar triggers of Aβ_1–40_ and Aβ_1–42_ aggregation, it can be assumed that antibodies specific to these forms of Aβ may have a preventive effect against the excessive formation of toxic Aβ oligomers in the brain without targeting intact Aβ and its physiological functions [139].

Based on the results of the studies described above, the role of autoantibodies to Aβ in the regulation of AD in humans remains ambiguous. It is also not known whether autoantibodies to more toxic and oligomerization-promoting Aβ forms, such as Iso-D7-Aβ and 3pE-Aβ, are formed in healthy adults and AD patients and, if so, whether they may influence the regulation of the disease.

#### 4.2.2. B Cells

While B cells are essential for the formation of autoantibodies to Aβ, their direct role in the pathogenesis of AD is rarely considered and remains a matter of debate. Initial studies have shown a decrease in the total number of circulating B cells in the peripheral blood of patients with AD, both in absolute numbers and as a percentage of peripheral blood mononuclear cells (PBMCs) [201,202]. On the other hand, an analysis of the B cell receptor (BCR) repertoire of B cells from the peripheral blood of AD patients demonstrated enrichment of some BCR clones as compared to healthy donors of the same age [54]. These results suggest that patients with AD have an accumulation of certain antigen-specific B cells involved in the immune response [54]. Based on the previously described accumulation of autoantibodies to the N-terminus of Aβ in a group of patients with more severe AD [127], a corresponding increase in the circulating memory B cells specific to the N-terminus of Aβ could be hypothesized. However, to date, the antigen-specificity of B cell clones accumulating in AD patients, as well as their specificity to various forms of Aβ, including the most common post-translational modifications of Aβ, is not known. In addition, the direct role of B cells in the pathogenesis of AD is unclear (Figure 1).

A recent study has shown that B cell deficiency in different mouse models of AD (3xTgAD, APP/PS1, 5xFAD) leads to a decrease in amyloid deposits at the base of the hippocampus and improved cognitive function [203] (Table 2). This study was conducted both in B cell-knocked-out mice (BKO) and in mice where B cells were transiently depleted by administration of antibodies to CD20 or B220 [203]. In the first case, B cell-deficient mice were obtained by crossing JHT mice that do not develop mature B cells [203] with 3xTgAD [204] or APP/PS1 [205] mice. Both B cell-deficient mouse models of AD had reduced amounts of amyloid deposits in the brain and improved cognitive abilities (at 50–60 weeks in 3xTgAD-BKO mice and 20–35 weeks in APP/PS1-BKO mice) as compared to animals of the same background in which B cells were not knocked out. In addition, α-CD20 or α-B220-mediated B cell elimination in 3xTgAD (60–70 weeks old) and 5xFAD (35–47 weeks old) mice showed reduced deposition of amyloid plaques in the brains of mice two months after the start of therapy. The results suggest that B cell elimination leads to the inhibition of AD and improved cognitive function. However, it is unclear what kind of B cells exacerbate disease progression in mouse models of AD, including their antigen specificity, mechanism of action, and the role they play in the brain parenchyma [203].In addition to autoantibody production, which may influence the development of AD in mice, several other possible mechanisms of B cell action may exacerbate AD development. One possibility is that autoantigen-specific B cells stimulate T cells, pathogenic for AD development, by presenting autoantigenic peptides (e.g., from Aβ) in complex with MHCs on their surface (Figure 1). The B cell-dependent activation of T cells can occur either distantly (in SLO) or locally (in the brain parenchyma) in the case of the colocalization of antigen-specific B and T cells. While the B cell-dependent activation of T cells in SLO is more likely due to scarce B cell presence in the brain, one study reported an increase in the numbers of B cells in the brain parenchyma of 5xFAD female mice [203]. In addition to the antigen-specific activation of cognate T cells, B cells may influence T cell responses in other ways [206,207]. B cells may also contribute to AD development through the hypersecretion of pro-inflammatory mediators, which, in turn, may contribute to the activation of microglia in the brain [19] and neuroinflammation (Figure 1).

In addition to stimulating the innate and T cell responses, B cells may also play a role in suppressing the inflammation and inhibiting the development of pathological processes. Some subpopulations of B cells (e.g., regulatory B cells) are known to express anti-inflammatory cytokines such as IL-10, IL-35, and TGFβ, which tend to limit the immune response and reduce the risk of autoimmune diseases [208]. Interestingly, it was recently shown that in 3-month-old 5xFAD mice, which had already begun to show AD pathology, IL-35 produced by B cells inhibited β-secretase (BACE1), which is responsible for the formation of β-amyloid from APP [209] (Figure 1). Thus, even in murine models of AD, B cells can play different, sometimes opposing, roles at different stages of disease progression. Further studies are required to address the role of B cells, including Aβ-specific B cells, in human AD pathology.

To summarize the above, B cells represent a heterogeneous population of cells with different antigen specificities, localizations, and functions [203]. An assessment of the contribution of B cells to the development of Alzheimer’s disease requires a detailed analysis of B cell subsets circulating in the blood and localizing into the brain at different stages of pathology development, and of their specificity to Aβ and other autoantigens.

## 5. Concluding Notes

In addition to the well-examined role of β-amyloids in AD pathogenesis, multiple studies have suggested the important role of the immune system in its regulation. However, at this time, there are outstanding gaps in the understanding of the immune response to Aβ and how it affects the predisposition of healthy adults to AD or progression of the ongoing disease. This includes insufficient information on (i) how various pathogenic forms of Aβ affect the innate immune system and inflammation, (ii) the involvement of various subsets of Aβ-specific CD8 and CD4 T cells in the regulation of the disease, (iii) the autoantigen-reactivity and function of B cell clones that accumulate in AD patients and their specificity to various forms of Aβ, (iv) the presence of autoantibodies to more toxic and oligomerization-promoting Aβ forms in healthy adults and AD patients and their role in the control of the disease, (v) the presence of local plasma cells in the brain parenchyma of AD patients, and (vi) the specificity of antibodies in the brain parenchyma to Aβ and other antigens. Resolving these questions is important for dissecting the interplay between various forms of Aβ and the immune system in the regulation of AD and could aid in the development of novel diagnostic and therapeutic tools for this debilitating disease.

## Figures and Tables

**Figure 1 cells-13-01624-f001:**
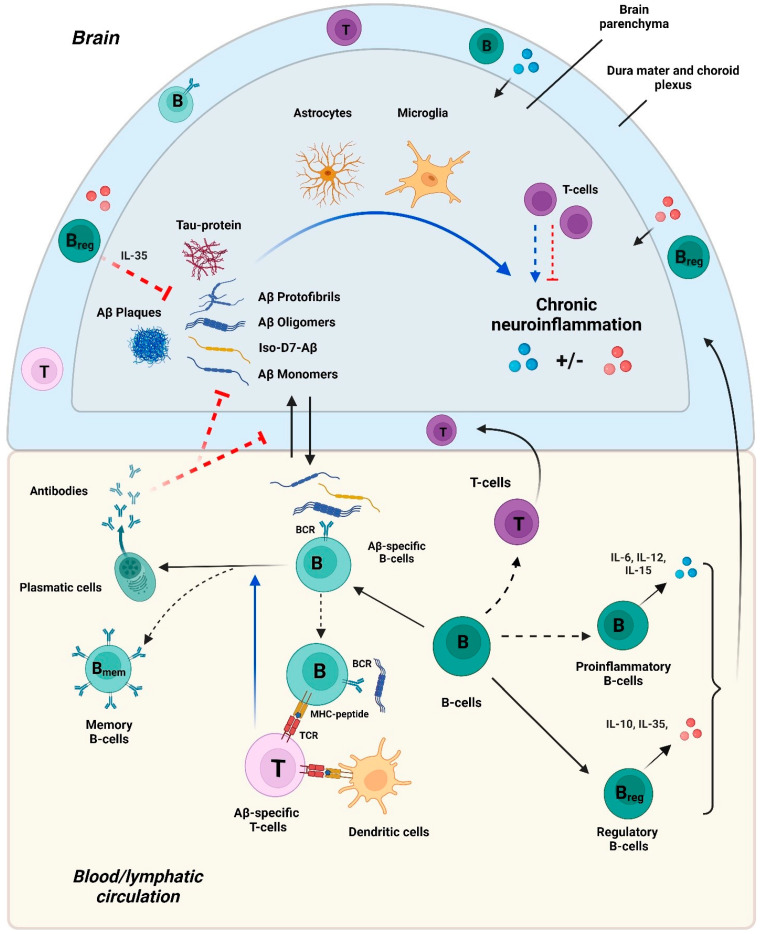
A model of Aβ and immune system participation in AD pathogenesis. **Notes:** blue arrows—stimulation, red arrows—inhibition; solid lines—established (acknowledged) regulatory pathways, dashed lines—potential (possible) regulatory pathways.

**Table 2 cells-13-01624-t002:** Mouse models of Alzheimer’s disease discussed in this review.

Mouse Model NR2B-Containing NMDA Receptor	Gene (Mutation)	Aβ Pathology	Tau Pathology	Neuronal Loss	Cognitive Impairment	First Described
PDAPP	APP (V717F)	6 mo. +	-	-	3 mo. +	1995 [140]
APP-Tg J20	APP (K670/671NL, V717F)	8 mo. +	-	3 mo. +	4 mo. +	[141]
APP/PS1	APP (KM670/671NL)	6 mo. +	-	8 mo. +	12 mo. +	2006 [142]
	PSEN1 (delta9)					
5xFAD	APP (KM670/671NL, I716V, V717I)	8 mo. +	-	6 mo. +	4 mo. +	2006 [143]
	PSEN1 (M146L, L286V)					
3xTgAD	APP (KM670/671NL)	6 mo. +	12 mo. +	unknown	4 mo. +	2003 [144]
	MAPT 0N4R (P301L)					
	Psen (M146V knock-in)					
SJL mice	expressing IFN-γ under the MBP promoter					[145]
APP/IFN-γ Tg	homozygous IFN-γ-Tg mice were bred with APP-Tg J20 mice					[146]

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
