# Peer review of "β-Amyloids and Immune Responses Associated with Alzheimer’s Disease"

_cells, 2024, doi:10.3390/cells13191624_

Round 1
Reviewer 1 Report
Comments and Suggestions for Authors
Alzheimer’s disease (AD) is associated with the accumulation of β-amyloids (Aβ) and the formation of Aβ fibrillar plaques in the brain. The lengths of Aβ peptides vary from 37 to 43 amino acid residues with Aβ1-42 most prone to aggregate.The authors suggest that insoluble Aβ aggregates are not likely to be the main cause for neurodegeneration which is more likely to arise from the large amounts of toxic soluble Aβ oligomers present intra- and extra-cellularly. Possible mechanisms of toxicity include disruption of cellular membranes, dysregulation of Na/K+ and Ca2+ion flows, pathological engagement with cell and mitochondrial membrane receptors, and induction of oxidative stress. Post-translational modifications can increase the tendency of Aβ to aggregate. The review then discusses the interaction of Aβ with the innate (microglia/astrocytes) and adaptive immune systems (B and T lymphocytes and neurtophils) in the pathogenesis of Alzheimer's disease. There is clear evidence of microglial astrocytic activation in all phases of AD which may initially be protective but later becomes toxic. The role of B and T cells is less clear, B cells can cross the blood-brain barrier (BBB) and become activated releasing antibodies and programming T-cells but reported findings have been inconsistent. The role of Aβ specific antibodies as an AD therapy is then debated - four have been licensed by the FDA but their efficacy is limited, they have adverse side effects, and disease progression is not halted. The review concludes by listing a number of knowledge gaps in the understanding the immune response to Aβ and how it affects the predisposition of healthy adults to AD or progression of the ongoing disease.
This is a well written and interesting review and it tabulates the mechanisms by which soluble Aβ oligomers are toxic, the various transgenic mouse AD models available for investigators and their merits and drawbacks. A model of Aβ and immune system participation in AD pathogenesis is presented in a figure. While this review is instructive, it rather leaves the reader with the impression that toxic soluble Aβ oligomers and the immune response to them are primarily responsible for causing AD and its progression. The review says little about tau pathology and that cortical tau tangles are only found when cortical Aβ deposits are present. Do the authors have a view on how rising Aβ levels might act to trigger tau hyper-phosphorylation which is the main killer of neurones?
Author Response
Reviewer 1:
Alzheimer’s disease (AD) is associated with the accumulation of β-amyloids (Aβ) and the formation of Aβ fibrillar plaques in the brain. The lengths of Aβ peptides vary from 37 to 43 amino acid residues with Aβ1-42 most prone to aggregate.The authors suggest that insoluble Aβ aggregates are not likely to be the main cause for neurodegeneration which is more likely to arise from the large amounts of toxic soluble Aβ oligomers present intra- and extra-cellularly. Possible mechanisms of toxicity include disruption of cellular membranes, dysregulation of Na/K+ and Ca2+ion flows, pathological engagement with cell and mitochondrial membrane receptors, and induction of oxidative stress. Post-translational modifications can increase the tendency of Aβ to aggregate. The review then discusses the interaction of Aβ with the innate (microglia/astrocytes) and adaptive immune systems (B and T lymphocytes and neurtophils) in the pathogenesis of Alzheimer's disease. There is clear evidence of microglial astrocytic activation in all phases of AD which may initially be protective but later becomes toxic. The role of B and T cells is less clear, B cells can cross the blood-brain barrier (BBB) and become activated releasing antibodies and programming T-cells but reported findings have been inconsistent. The role of Aβ specific antibodies as an AD therapy is then debated - four have been licensed by the FDA but their efficacy is limited, they have adverse side effects, and disease progression is not halted. The review concludes by listing a number of knowledge gaps in the understanding the immune response to Aβ and how it affects the predisposition of healthy adults to AD or progression of the ongoing disease.
This is a well written and interesting review and it tabulates the mechanisms by which soluble Aβ oligomers are toxic, the various transgenic mouse AD models available for investigators and their merits and drawbacks. A model of Aβ and immune system participation in AD pathogenesis is presented in a figure. While this review is instructive, it rather leaves the reader with the impression that toxic soluble Aβ oligomers and the immune response to them are primarily responsible for causing AD and its progression. The review says little about tau pathology and that cortical tau tangles are only found when cortical Aβ deposits are present. Do the authors have a view on how rising Aβ levels might act to trigger tau hyper-phosphorylation which is the main killer of neurones?
Response to reviewer:
- We thank the reviewer for this important point! To address it we have included a new paragraph that discusses our team’s view on the possible link between rising Aβ levels and tau hyper-phosphorylation in AD:
2.1.5 AD-related target protein phosphorylation
pages 7/8, lines 183-200
Reviewer 2 Report
Comments and Suggestions for Authors
Kolobova E.et al reviews the association between Amyloid beta and the immune system in a clear and well-organized manner. The structure is straightforward, making the complex topic easier to understand. Additionally, the review includes comprehensive references that enhance its credibility and depth. The paper could be considered to publish if following comments were addressed:
1. The evidence regarding T cells presented in section 4.1 heavily relies on transgenic models that artificially overexpress Amyloid beta in the brain. Given this artificial overexpression, it is expected to observe an activated immune response, as it represents a significant deviation from normal physiology. However, drawing conclusions about the pathological contribution of T cells based on these findings from such an artificial model raises significant concerns.
2. Similar to B cells. It will be important to include knock-in models to investigate the immune response of AD. The only summary of transgenic models is insufficient.
3. The primary role of immune response in the brain is made by astrocyte and microglia and some other glia cells, however, the section for the function of these cell types is very limited in this review.
4. How about other immune cells? Such NK cells?
Author Response
Reviewer 2
Kolobova E.et al reviews the association between Amyloid beta and the immune system in a clear and well-organized manner. The structure is straightforward, making the complex topic easier to understand. Additionally, the review includes comprehensive references that enhance its credibility and depth. The paper could be considered to publish if following comments were addressed:
- The evidence regarding T cells presented in section 4.1 heavily relies on transgenic models that artificially overexpress Amyloid beta in the brain. Given this artificial overexpression, it is expected to observe an activated immune response, as it represents a significant deviation from normal physiology. However, drawing conclusions about the pathological contribution of T cells based on these findings from such an artificial model raises significant concerns.
Response:
We completely agree with the reviewer’s point of view that animal model-related findings may not be relevant to human disease. Therefore, we made an effort to separately discuss the findings in mice and in humans in the text. Moreover, in the modified manuscript we have included the following sentence into discussion:
“In addition, whether mouse-models’ derived conclusions about Aβ-specific T cells may be relevant to human AD pathology remains an opened question.” p. 13, lines 358-359
- Similar to B cells. It will be important to include knock-in models to investigate the immune response of AD. The only summary of transgenic models is insufficient.
Response:
The same applies to discussion of B cells in mice and humans and to our conclusions. Please see p.19, lines 529-530
- The primary role of immune response in the brain is made by astrocyte and microglia and some other glia cells, however, the section for the function of these cell types is very limited in this review.
Response:
Because multiple excellent reviews (referenced in the text) have been dedicated to the innate immune response in the brain in AD, in this review we have attempted to focus more on the adaptive immune response, especially on B cells (as laid out in the abstract and the introduction).
- How about other immune cells? Such NK cells?
Response:
-We thank the reviewer for this suggestion! We have included a paragraph discussing possible contribution of NK cells to the disregulation/treatment relevant to the AD in 3. p.10, lines 272-282